# Correlation between Spatio-Temporal Evolution of Habitat Quality and Human Activity Intensity in Typical Mountain Cities: A Case Study of Guiyang City, China

**DOI:** 10.3390/ijerph192114294

**Published:** 2022-11-01

**Authors:** Yongfei Luan, Guohe Huang, Guanghui Zheng, Yuee Wang

**Affiliations:** 1Ministry of Education, Key Laboratory of Regional Energy and Environmental Systems Optimization, Environmental Research Academy, North China Electric Power University, Beijing 102206, China; 2School of Environment, Beijing Normal University, Beijing 100875, China; 3School of Architecture and Art, Central South University, Changsha 410083, China; 4Guizhou Institute of Light Industry, Guiyang 550025, China

**Keywords:** land-use change, habitat quality, mountainous cities, human activity intensity, Guiyang City

## Abstract

The acceleration of the urbanization process brings about the expansion of urban land use, while changes in land-use transformation affect the urban habitat quality, and land-use change brings a threat to regional sustainable development. Against such a backdrop, the assessment of land use on the habitat quality and the relationship between the intensity of human activities is becoming a hot spot in terms of the current land use coordinated with habitat quality. Based on the land-use data of Guiyang in 2000, 2005, 2010, 2015 and 2020, the spatial–temporal evolution characteristics of habitat quality in the study area, combined with the spatial correlation between human activity intensity and habitat quality, were hereby analyzed using the InVEST model. The impact of human activity intensity on habitat quality was correspondingly analyzed. The results show that: (1) From 2000 to 2020, the habitat quality level in Guiyang remained stable without drastic changes, but the changes showed hierarchical distribution and were scattered, mainly reflected in the urban expansion areas of the urban–rural fringe and the key areas of industrial development, and the ecological environment quality fluctuated in a small range. (2) From 2000 to 2020, the intensity of human activities in Guiyang was mainly affected by the relatively concentrated distribution, featuring obvious and significant changes. From 2010 to 2015, the high-impact area surrounded the Guanshan Lake New Area, and the regional habitat quality presented a downward trend. In 2020, the high-impact area of the main urban area and the key industrial development zone was expected to be formed, while the low-impact area was still distributed in forest areas with complex natural conditions. (3) From 2000 to 2020, there was a significant positive correlation between human activity intensity and habitat quality in Guiyang, and such a spatial correlation was weak from 2000 to 2005. The period from 2015 to 2020 witnessed the rapid development of urban construction in Guiyang, human construction activities continue to affect the urban habitat quality. The results show that the intensity of human activities on the promoting function of land use, and the dependencies between them should be considered at the same time, and that explorations on the influence of human activities on land-use intensity and habitat quality of space link are crucial to improving the efficiency of urban land use and ecological environment protection, as well as the coordination between land use and the sustainability of urban development.

## 1. Introduction

With the acceleration of the urban industrialization and urbanization process, as well as the constant enhancement of the scale and intensity of human transformation of natural resources, human economic activities have been continuously exerting a certain impact on the function, structure and space of the regional ecological environment. At the same time, due to the unique natural attributes and complexity of the mountain structure, the blindness of urban development and construction has led to the deterioration of the ecological environment and the frequent occurrence of secondary disasters, such as geological disasters in mountain cities, which seriously restricts the healthy development of cities and the quality of habitats [1,2]. Based on such a background, assessment of the correlation between the intensity of human activities and the spatial–temporal changes in habitat quality has become a hot issue in the research field in the case of evaluating the health and habitat quality level in mountain cities [3,4].

The types of geological structure in mountainous cities affect the layout of urban construction, while urban expansion and change affect the change of the types of regional land resource structure, thereby leading to the transformation of urban cultivated land, grassland, woodland and construction land, and bringing a certain degree of negative impact on the fragmentation and sustainability of the ecological landscape pattern [5]. Urban expansion and change increase the demand for land resources. Land is the basis of urban development, and affects the change of urban ecological environment and land-use map spots, which also results in the change of the land-use spatial pattern. In this case, analyzing the impact of spatio-temporal changes in land use on habitat quality and exploring the correlation characteristics between changes in human activity intensity and habitat quality are endowed with an important reference value for the ecological protection in urban areas and the healthy development in mountainous cities [6]. Based on the previous study of the impact of land use change on habitat quality, scholars have carried out several assessment studies from different perspectives, scales, and regions combined with diversified methods. In the early studies, habitat quality changes of wildlife habitats were evaluated mainly through biodiversity and habitat changes [7]. This method is time-consuming and laborious, affected by the carrying capacity of the natural environment, and subject to a strong subjectivity and limited regional conditions, making it difficult to carry out investigations on different scales. Then, the InVEST model was generated to better simulate the changes in land ecological service quality under different land cover backgrounds. The qualitative and quantitative methods were used to evaluate the changes in habitat quality, thereby realizing the spatial and visual expression of habitat quality function assessment, and vividly describing the spatial–temporal variation characteristics of habitat quality. It provides a reference basis for decision makers to evaluate the benefit and impact of human activity intensity. The advantages of fewer data requirements and high simulation accuracy of the InVEST model have made it widely used in the field of habitat quality assessment [8]. At present, both domestic and foreign scholars have used the assessment model for the ecological service value, NPP and NDVI habitat index evaluation, geographic detector, human activity intensity and geographic regression model to quantitatively evaluate the spatial change and impact of habitat quality [9,10,11,12]. However, there are relatively few studies on assessing habitat quality changes in mountainous cities using the InVEST model, and few studies are conducted on the correlation with human activity intensity. To this end, evaluating the impact of land use on habitat quality using quantitative methods has become one of the hot issues in the qualitative assessment of habitat quality change [13], and the application of land-use change data to the analysis on regional habitat quality changes is of great practical significance for the qualitative assessment of urban habitat quality and sustainability. 

Guiyang is an innovative city in southwest China, also a typical karst landform region city, and most of the cities are mountainous landforms. In this case, the influencing factors of human activity intensity on habitat quality were hereby discussed, and the correlation between them was analyzed by taking Guiyang city as an example. The results possess certain representativeness and typicality for the development and health quality assessment of mountain cities in southwest China. Based on the land-use change data of 2000, 2005, 2010, 2015 and 2020, the interaction between habitat quality evolution and regional human activity intensity in Guiyang was hereby explored, which is endowed with important research significance for the development of ecological environment quality in regional mountain cities. Research on the land-use change of time and space on the effect of land environment quality was conducted, and the results show a correlation between the two assumptions. Quantitative evaluation of the habitat quality change of Guiyang City from 2000 to 2020 was conducted by virtue of the InVEST model and through many index map overlay analysis human activity intensity index (HAI). Finally, the bivariate spatial autocorrelation and geographically weighted regression model methods were used to explore the impact of human activity intensity on habitat quality and its correlation, so as to provide a reference for the study of urban construction and ecological environment quality in Guiyang, as well as the ecological environment and sustainability of mountainous cities in southwest China. 

## 2. Data Material Sources and Research Methods

### 2.1. Overview of the Study Area

Guiyang City is located in the middle part of the original hills of the central Guizhou Mountains, which belongs to the watershed zone of the Yangtze River and the Pearl River. The terrain is high in the southwest and low in the northeast. The highest elevation of the province is 2885 m and the lowest is 152 m. The geomorphology here mainly consists of mountainous and hilly areas (Figure 1), among which, the mountain area covers about 4217 km^2^, accounting for 52.43%, while the hill area is about 2840 km^2^, taking up 35.31%, and other types of land account for about 12.26%. By the end of 2020, the total population was 5.9898 million, the primary industry value was USD 0.0025 billion, that of the secondary industry was USD 0.0214 billion, that of the tertiary industry was USD 0.0355 billion, the per capita GDP was USD 9940.76, and the total financial revenue was USD 12,136.9624 billion. The total land area of the city is 8043 km^2^ in the year 2022, and the land type area is taken as an example. Among them, the cultivated land area covers an area of 2112.49 km^2^, accounting for 26.26%; the forestland area is 3906.57 km^2^, accounting for 48.57%; the grassland area is 1342.58 km^2^, accounting for 16.69%; the water body area is 135.19 km^2^, taking up 1.68%; the construction land area covers 542.36 km^2^, accounting for 6.74%; and the unutilized land is 3.84 km^2^, accounting for 0.047%.

### 2.2. Data Sources

In this paper, the land-use change data were downloaded from the Resources and Environment Data Sharing Center of Chinese Academy of Sciences (http://www.resdc.cn), (accessed on 1 January 2020). After cutting and splicing, the land-use data of Guiyang City were intercepted. The spatial resolution of the data is the 30 m × 30 m data type, and the data accuracy can reach more than 90% after testing. The data included the economic development data from the statistical yearbook published by the People’s Government of Guiyang (http://www.guiyang.gov.cn), (accessed on 1 January 2020), the data in 2020 economic development in 2021 Guiyang City from statistical yearbook data, and DEM elevation data of the spatial resolution of 30 m. The base map derived from the geographic information public service platform (https://guizhou.tianditu.gov.cn), (accessed on 1 January 2020) in the administrative division scope of data in Guizhou, Guiyang City on the basis of Guiyang City in 2021 (not including Guian new district).

### 2.3. Research Framework

The research is based on the land use change data for 2000, 2005, 2010, 2015 and 2020. Firstly, the research data and DEM data were collected for preprocessing. After vectorization, accuracy testing and verification procedures were carried out. Secondly, for land data and statistical yearbook data, the characteristics of land spatial change were described using the transfer matrix, atlas analysis, InVEST model and other methods. Thirdly, the human activity intensity index and bivariate spatial autocorrelation were adopted to analyze the closeness and influence of their spatial connection. Finally, the spatial characteristics based on the characteristics of land-use change and habitat quality change in these five periods were summarized, and the response trend of habitat quality change was analyzed. The research framework was shown in Figure 2.

## 3. Research Methods

### 3.1. Habitat Quality

Land change is the most direct factor affecting habitat quality. Through the habitat quality module in the InVEST model, it evaluates the habitat quality of the region. This module combines the regional landscape type information, and evaluates the sensitive sources of threat factors using threat factors and land-use types on the basis of land-use change data, and is generally used to represent the characteristics of regional habitat quality change [14]. The calculation formula is as follows:(1)$$Q_{xj=H_{j}}\left(1-\frac{D_{xj}^{z}}{D_{xj}^{z}+K^{Z}}\right)\;$$
where *Q_xj_* represents the habitat quality index of raster *x* in land type *j*; *H_j_*, the habitat suitability of type *j* land types; *D_xj_*, the threat degree of raster *x* in the land type *j*; *K* is half satiety constant; and *z* is constant 2.5. The formula of the threat degree is:(2)$$D_{xj}=\overset{R}{\underset{r=1}{\sum }}\sum_{y=1}^{Y_{r}}\left(\frac{w_{r}}{\sum_{r=1}^{R}w_{r}}\right)r_{y}i_{rxy}\beta_{x}S_{jr}\;\;\;$$
where *R* denotes the threat factor; *y* is the total number of r threat grids; *Y_r_* is the number of a set of threat grids in *r* threat factors; *W_r_* is the weight of threat factor; *r_y_* is the threat factor value of grid *y*; *i_rxy_* is the threat level of grid *x* of threat factor *r_y_* of threat grid y; *β_x_* is the legal protection level of grid cell *x*; *S_jr_*, the sensitivity degree of land type *j* to threat factor *r*; *D_xy_* is the straight-line distance between grid *x* and *y*; and *Drmax* is the maximum distance of threat factor *r*. The linear formula of *i_rxy_* can be expressed as:(3)$$i_{rxy}=1-\left(\frac{d_{xy}}{d_{rmax}}\right)\left(\;\mathrm{Linear}\;\mathrm{distance}\;\mathrm{decay}\;\mathrm{function}\right)$$
(4)$$i_{rxy}=exp\left(-\left(\frac{2.99}{d_{\;rmax}}\right)d_{xy}\right)\left(\;\mathrm{Exponential}\;\mathrm{distance}\;\mathrm{decay}\;\mathrm{function}\right)$$

According to the existing results of urban habitat quality research and the actual situation of Guiyang, the reference value range given by the model was determined. Zhou T, He J and Liu J [15,16,17] selected the cultivated land, the construction land and the unutilized land as threat factors in relevant studies, and consulted experts in related fields. The sources and weights of habitat quality threats (Table 1) and the relative sensitivity of habitat suitability and threat sources (Table 2) in Guiyang were thus formulated.

### 3.2. Human Activity Intensity

Human activity intensity is an important driving factor affecting regional habitat quality change. Quantitative assessment of human activity intensity is the basis for analyzing ecosystem stability. The index model for human activity intensity was hereby used to quantitatively describe the impact of regional ecosystem change, and then to evaluate the relationship between human activities and land-use change. The index model for human activity intensity, which can describe the impact of human activity intensity on the ecosystem, was selected for evaluation [18], and its calculation formula is as follows:(5)$$\mathrm{HAI}=\sum_{i=1}^{n}\frac{A_{i}P_{i}}{\mathrm{TA}}$$
where HAI stands for human activity intensity index; n is the number of land types; A_i_ is the area of Class i land type; P_i_ is the intensity coefficient of human activities of type i ecological value; and TA is the total area.

According to the existing research results [19], the coefficient table for human activity intensity (Table 3) of different land types was determined for calculation, and the value of the human activity intensity index in the unutilized land was taken as the reference [20]. According to the calculation results, the impact types were divided into five grades, i.e., low HAI ≤ 0.2, low 0.2 < HAI ≤ 0.4, medium 0.4 < HAI ≤ 0.6, high 0.6 < HAI ≤ 0.8 and high 0.8 < HAI [20].

### 3.3. Bivariate Autocorrelation Analysis

The spatial distribution characteristics and aggregation degree of factor attributes were explored, and spatial correlation and tests were conducted through global and local spatial autocorrelation. The global Moran’s I index verifies the spatial agglomeration trend of relevant attributes in the region [21], and the calculation formula is as follows:(6)$$I=n\overset{n}{\underset{i=1}{\sum }}\overset{n}{\underset{j=1}{\sum }}W_{ij}\left(X_{i}-\bar{X}\right)\left(X_{j}-\bar{X}\right)/\overset{n}{\underset{i=1}{\sum }}\overset{n}{\underset{j=1}{\sum }}W_{ij}\overset{n}{\underset{i=1}{\sum }}{\left(X_{i}-\bar{X}\right)}^{2}\;$$
where *X_i_* and *X_j_* are the observed values of the elements in the *i* and *j* regions; and *W_ij_*, the weight matrix of the *i* and *j* spatial positions. When *i* and *j* are adjacent, *W_ij_* = 1; otherwise, *W_ij_* = 0. The global Moran’s index *I* is between (−1, 1), and Moran’s index value is positive, suggesting that the spatial autocorrelation is spatially clustered; otherwise, the spatial distribution tends to be scattered when the elements are significantly different; and the random distributions are irrelevant when Moran’s index is 0.

The bivariate spatial autocorrelation method was used by Anselin et al., to analyze the spatial relationship between human activities and habitat quality [22]. The formula of Moran’s I index can be expressed as:(7)$$I_{kl}^{i}=z_{k}^{i}\;\overset{n}{\underset{j=1}{\sum }}w_{ij}\;z_{l}^{j}\;\;\;$$
where *W_ij_* stands for *i* and *j* space position weight matrix, respectively; $Z_{k\;}^{i}=\frac{x_{k}^{i}-{\bar{x}}_{k}}{k}\;Z_{l}^{i}=\frac{X_{l}^{i}-{\bar{x}}_{l}}{l}$, ${\bar{x}}_{k}$*,*
${\bar{x}}_{l}$, is the average of the properties of *K* and *L*; and $x_{k}^{i}$*,*
$X_{l}^{i}$ are the values of *i* attribute *k* and *L*, respectively;

According to the calculation of Local Moran’s I index, the regions of the birth environment quality and the human activity type were divided into four types of human activity and habitat quality types: high high, low high, low low and high low.

### 3.4. Analysis of Geographically Weighted Regression Model

The geographically weighted regression model is a spatial analysis technique for parameter estimation, which is based on the establishment of a traditional regression model (OLS), and can simulate the spatial non-stationarity of different geographic spaces and verify the influence of different geospatial variables on regions [23]. The calculation formula can be expressed as:(8)$$y_{i}=\beta_{0}\left(u_{i}+v_{i}\right)+\sum_{k}\beta_{K}\left(u_{i}+v_{i}\right)\;x_{\;ik}+\varepsilon_{i}\;\;\;\;\;$$
where $y_{i}$ is the influence value of variable regression; (*u_i_*, *v_i_*) is the geographic coordinates of *i* samples; *x_ik_* is the value of the *k* independent variable in the *i* sample unit; *k* is the number of independent variables; *i* is the number of sample units; $\;\varepsilon_{i}$ is a random interference term; and *β_k_*(*u_i_*, *v_i_*), the unit value of continuous function *β_k_*(*u*, *v*) in sample *i*.

## 4. Results and Analysis 

### 4.1. Spatial Evolution Characteristics of Habitat Quality

Through the habitat quality module of the InVEST model, the habitat quality level area and change ratio table of Guiyang (Table 4) and the habitat quality spatial change distribution map of Guiyang from 2000 to 2020 (Figure 3) were obtained. Referring to previous research results [16], the results of habitat quality assessment in Guiyang were divided into five categories, i.e., low (0–0.2), relatively low (0.2–0.4), moderate (0.4–0.6), relatively high (0.6–0.8) and high (0.8–1), according to the Equal Interval method in ArcMap using ArcGIS10.3 software.

From the time scale perspective, due to the acceleration of urbanization construction, habitat quality changed around the urban core area, and small range fluctuation was observed from 2000 to 2010. From 2010 to 2020, the demand for the land to be used for industrialization and urban construction in Guiyang increased continuously, resulting in a small range of fluctuations in the land map spots in non-central urban areas. Areas with more obvious changes included Guanshanhu District, Baiyun District, Huaxi District, Zazuo Town, Zhandjie Town and other areas with significantly reduced habitat quality.

From the spatial scale perspective, the habitat quality grade in Guiyang mainly belonged to the low and high categories, which was relatively concentrated on the whole. The variation characteristics of the habitat quality area from 2000 to 2020 were 2565.49 km^2^ in 2000, accounting for 31.90%, and 2487.48 km^2^ in 2020, taking up 30.93%. The area with low habitat quality was 0.97% in space, and the decline rate was rather limited. In 2000, the area with high habitat quality in Guiyang was 3049.34 km^2^, 37.91%, which was changed to 2707.99 km^2^ in 2020, accounting for 33.67%. The area with high habitat quality decreased to 341.35 km^2^, accounting for 4.24%. Mainly affected by urbanization and industrialization, some forest resources and ecological land were destroyed, thereby resulting in a gradual decline in habitat quality.

In general, the habitat quality in Guiyang was good and relatively stable from 2000 to 2020, and there were few areas with large fluctuations. However, small area fluctuations were found in the core areas of economic development and urban expansion areas such as Yanshanhong Town, Zazuo Town and Zhanjie Town in the suburban area of the study area.

### 4.2. Spatio-Temporal Characteristics of Human Activity Intensity

ArcGIS10.3 software was used to evaluate the driving factors of the impact of regional habitat quality, and analyze the ecosystem stability by the change of human activity intensity, and the human activity intensity index was calculated by the 1 km × 1 km unit. The human activity intensity index of Guiyang in the five periods of 2000, 2005, 2010, 2015 and 2020 was obtained (Figure 4). The intensity of human activities in Guiyang from 2000 to 2020 in the study area was dominated by a relatively concentrated distribution of impacts, with obvious changes and significant differences found in the spatial impacts. HAI 2000 high concentration distribution in Yunyan District and the main Nanming portions characterized and Baiyun District, the main reason is that the main population centralization degree is higher, and is related to the height of the middle part of the industrial concentration. Low activity areas are mainly concentrated in Kaiyang, the map of Xifeng County, a remote area of the region, and the forest coverage rate is higher. In 2005, the influence area of HAI increased significantly, and the high activity influence mainly extended outward around the central city, forming the structure layout of an echelon encircle with the high influence part. From 2010 to 2015, the high-impact human activity space moved to Guanshan Lake District, and some exurb counties such as Chengguan Town in Kaiyang County were also subject to a certain impact. In 2020, the human activity intensity space in Guiyang formed a high-impact area represented by Yunyan District, Nanming District and Huaxi District. The human activity intensity in this area had a great impact on regional habitat quality, showing a positive correlation between human activity intensity and habitat quality. At the same time, the counties in Kaiyang, Xiuwen and Xifeng counties and the towns with better economic development showed a high spatial pattern influenced by the intensity of human activities. In addition, the areas with low impact of human activities in 2020 were distributed in areas with high forest coverage and complex topography, which had less impact of human activities and relatively high habitat quality.

### 4.3. Bivariate Spatial Correlation of Human Activity Intensity

The spatial autocorrelation index between human activity intensity and habitat quality in Guiyang from 2000 to 2020 was analyzed using Geoda 095i software, and the scatter plot of Moran’s I from 2000 to 2020 was obtained (Figure 5). As shown in the figure, the distribution of Moran’s I is relatively uniform in all quadrants, with more distributed in the first quadrant, indicating the obvious spatial correlation characteristics of the intensity of human activities and habitat quality space. The scatter analysis of Moran’s I trend shows that the intensity of human activities and habitat quality space in the study area are correlated, and that the correlation is relatively significant. The correlation between the two periods from 2000 to 2005 was weak, presenting a gradually weakening trend. The first quadrant of 2010 showed a relatively obvious positive correlation, and the change from 2015 to 2020 was relatively significant, indicating a negative correlation. The spatial correlation between the intensity of human activities and habitat quality was weak from 2000 to 2005. Considering the constraints of economic development and the topographic conditions, less amount of land was used for ecological land transfer and construction in Guiyang during this period, and the impact of human economic activities on urban habitat quality was weak as well. In 2010, due to the implementation of the policy of returning farmland to forest or grassland, the regional habitat quality was improved to a certain extent. However, the habitat quality in Guiyang decreased significantly from 2015 to 2020, which was attributed to the upsurge of urban construction in Guiyang, and the intensity of human activities affected the regional habitat quality level. The transformation of a large number of ecological land and water body areas into construction land gave rise to the relatively obvious and frequent spatial land-use map changes. The spatial differentiation of the regions with correlation in Moran’s scatter plot was analyzed using the LISA cluster analyzing method, and the LISA cluster map was drawn by the Z test (P = 0.05) (Figure 6). The relationship between human activity intensity and habitat quality was significant, but the high-high and the low-low types showed a clustering trend. From 2000 to 2020, the proportion of high-high agglomeration distribution areas presented a trend of gradual decrease, and the decrease was not obvious. The significance level of LISA cluster analysis shows that most areas of Guiyang are not significant, while the high-high type showed a high significance level that was staggered in the 0.01 and the 0.05 region. From 2000 to 2020, the distribution with a significance level of 0.01 showed a dynamic change. In 2010, the area of 0.01 distribution decreased, representing an increase in the spatial difference, while the area of 0.05 distribution increased, indicating that the spatial difference was being gradually narrowed.

### 4.4. Spatial Variation of the Impact of Human Activity Intensity on Habitat Quality

The least square model and geographically weighted regression model were analyzed using ArcGis10.3 software, and the AICc values were −5063.480 and −8781.726, respectively. When the AICc value of the least square method and the geographically weighted regression model is greater than 3, the results of the geographically weighted regression simulation are more reasonable [23]. In this study, the difference between the two is 3718.246, indicating a better result of geographically weighted regression than that of the least square model. The R2 of the geographically weighted regression model increased to 0.410, further indicating the favorability and suitability of the simulation results of the geographically weighted regression for this study (Figure 7).

From the time scale perspective, the human activity intensity index exercised a significant impact on habitat quality in Guiyang from 2000 to 2020, with the negatively affected area covering an area of 21.52% in 2000, 16.95% in 2015 and 18.32% in 2020, presenting dynamic change characteristics and an unstable trend. From 2000 to 2020, there was a positive correlation between the surrounding area of Guiyang and the area with a high forest coverage rate. The habitat quality in this area, such as Huaxi District and Xiuwen County, maintained a favorable trend due to the regional land nature. The areas with the negative impact of human activity intensity and habitat quality were mainly distributed in the main urban areas and key areas of economic development, where the regional habitat quality was declining due to urban construction and other reasons.

From the spatial scale perspective, the intensity of human activities on the spatial difference of habitat had a significant difference in the quality and performance as the impact significant of the Guiyang City core area, while the impact of the city’s surrounding areas is weak. Most of the core areas of the main city are negative value areas, while the marginal areas and forest areas are positive value areas. Based on this phenomenon, shows that the impact of human activity intensity on habitat quality mainly was negative. It is mainly caused by the restriction of territorial space planning and the influence of regional natural conditions. Especially in recent years, the development and construction of the new city in Guanshanhu District have been greatly accelerated, resulting in the intensification of human activities, which exerts a significant impact on the local regional habitat quality. In this case, attention should be paid to the harmonious relationship between habitat quality for the local land use and development, and the promotion of the sustainability of regional habitat quality development.

## 5. Discussion

Previous studies on the quality of human settlements in Guiyang are mainly based on the evaluation of a single habitat quality unit in the city, and the lack of comprehensive consideration of the spatio-temporal coupling relationship between habitat quality changes and human activity intensity. Based on the land-use change data of Guiyang from 2000 to 2020, the spatial and temporal characteristics of habitat quality changes in Guiyang were hereby analyzed using the habitat quality module of InVEST model. At the same time, the human activity intensity index, bivariate autocorrelation method and geographically weighted regression model were used to analyze the impact and spatial correlation characteristics of habitat quality, and to evaluate the spatial and temporal changes of habitat quality evolution. The results are endowed with a reference value and practical significance for the analysis of factors affecting the development, construction and habitat quality change of similar cities in the study area.

Habitat quality is an index reflecting regional ecological environment change, while land use change is an important factor affecting habitat quality change. Urban land use and development lead to the continuous expansion of the scale and scope of land use, but also have a certain impact on the quality of the ecological environment. Land-use transfer and change, with cultivated land, grassland and forestland converted into construction land as the main type, jointly form the habitat quality change pattern of “urban and rural areas and key industrial development areas”, which also confirms that urban development and industrial construction affect habitat quality change [14,24,25,26]. 

The impact of human activities intensity on habitat quality presents a spatial correlation, indicating that the intensity of human activities in the study area affects the habitat quality change, and the impact of population aggregation, economic development and policies make the changes mainly concentrated on the central urban area and the densely populated areas. From 2015 to 2020, the quality of urban habitat in Guiyang showed a continuously declining trend, which is consistent with the rapid urban development period of Guiyang [27]. 

Guiyang City is a typical karst mountain city in southwestern China, and is provided with unique ecological resource advantages in the process of urban construction and development, but is also exposed to the problem of inefficient land utilization, affected by the natural factors in the process of relatively concentrated land use, and driven by the economic driver and inappropriate land development that present the declining trend in urban habitat quality, which also brings certain pressure to the urban planning and management, and future land use should take into consideration the market economy and administrative intervention. The development models of ecological quality are favorable, regulating the functions of urban land examination and approval, the intensive economical utilization of land, as well as the city and the habitat quality of balanced and sustainable economic development. 

However, limited by the sources and methods of data, the study is still subject to problems concerning the selection of variables, the accuracy of land-use data, and the selection of sensitive sources, threat factors and weights of the InVEST model. Although it refers to the existing literature, the selection of data is still subjective to some extent. In future research, more qualitative and quantitative evaluation methods should be integrated into the research for higher-level data accuracy, for the better selection of index factors, and also for the improvement of the research methods, so as to provide more data and method support for the research. It is expected that the research results can be of practical significance and guidance for the coordination of urban economic development and ecological environment.

## 6. Conclusions

Land change is the most direct factor affecting habitat quality. This paper innovatively proposes a comprehensive regional factor index for evaluating urban habitat quality assessment and land use. This index comprehensively considers the correlation between habitat quality change and human activity intensity, and reveals the spatio-temporal characteristics of land use, human activity intensity and urban spatial pattern evolution. We evaluated the impact of land use change on habitat quality and the coordination between them. Based on the data on land use change in 2000, 2010 and 2020, the spatial and temporal characteristics of urban development and the interaction between land use were analyzed, and urban development was identified as a better level for the promotion of habitat quality, thereby providing references and suggestions for the improvement of relevant problems. Based on the above ideas and methods, the conclusions can be drawn as follows:
(1)From 2000 to 2020, the habitat quality level in Guiyang remained stable without drastic changes, but the changes showed a hierarchical and scattered distribution, mainly reflected in the urban expansion areas of the urban–rural fringe and the key areas of industrial development, and the ecological environment quality fluctuated in a small range. (2)From 2000 to 2020, the intensity of human activities in Guiyang was mainly affected by the relatively concentrated distribution, presenting obvious and significant changes. From 2010 to 2015, the high-impact area surrounded the Guanshan Lake New Area, and the regional habitat quality showed a downward trend. In 2020, the high-impact area of the main urban area and key industrial development zone was formed, while the low-impact area was still distributed in forest areas with complex natural conditions, which was less affected by the intensity of human activities. (3)From 2000 to 2020, there was a significantly negative correlation between human activity intensity and habitat quality in Guiyang. The spatial correlation between the intensity of human activities and habitat quality was weak from 2000 to 2005. Considering the constraints of economic development and the topographic conditions, less amount of land was used for ecological land transfer and construction in Guiyang during this period, and the impact of human economic activities on urban habitat quality was weak as well. The period from 2015 to 2020 is a period featuring the rapid development of urban construction in Guiyang, when human construction activities continued to affect the urban habitat quality, and the land use map spots changed frequently and obviously. The land use change is the main reason for the habitat quality change.(4)Limitations of the Study. Some limitations exist in our study. For example, land-use change is an uncertain and dynamic process. Due to the heavy workload of data processing and the difficulty of data collection, the data used in this study covers the period from 2000 to 2020. In future research, data from more stages can be obtained for comparison, so as to explore the spatio-temporal evolution law of human settlement’s environment quality and its influencing factors from more micro levels. The study can be enriched by obtaining air pollution volatility indices and other natural factors in certain sectors of the study area. This will be the focus of our future research.

## Figures and Tables

**Figure 1 ijerph-19-14294-f001:**
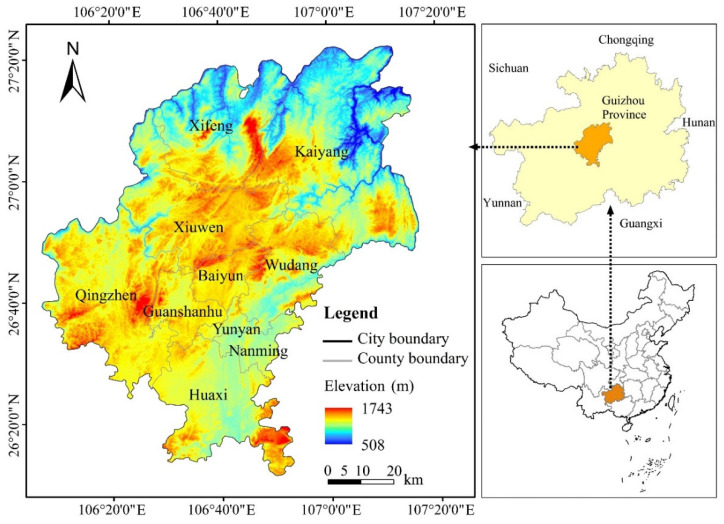
Geographic location and elevation of Guiyang City.

**Figure 2 ijerph-19-14294-f002:**
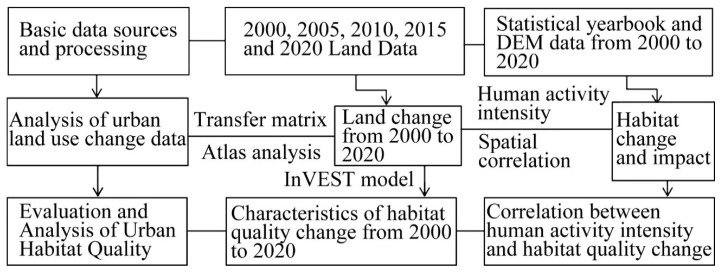
Research framework.

**Figure 3 ijerph-19-14294-f003:**
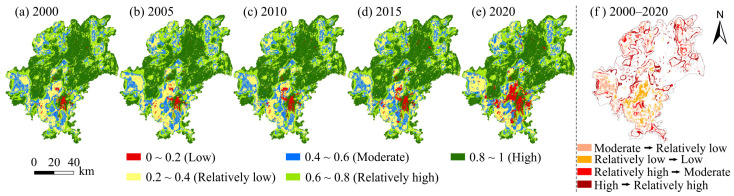
Spatial variation of habitat quality in Guiyang from 2000 to 2020.

**Figure 4 ijerph-19-14294-f004:**
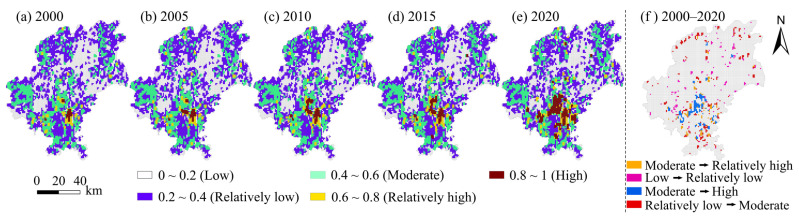
Spatial distribution of human activity intensity in Guiyang from 2000 to 2020.

**Figure 5 ijerph-19-14294-f005:**
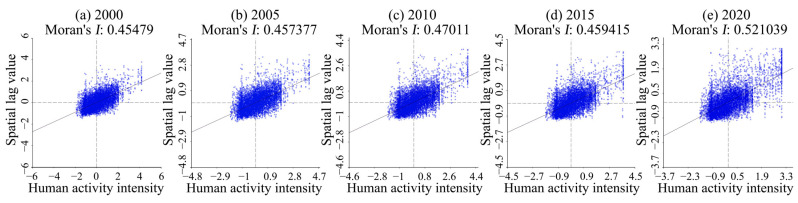
Moran’s scatter plot of changes in human activity intensity and habitat quality in Guiyang from 2000 to 2020.

**Figure 6 ijerph-19-14294-f006:**
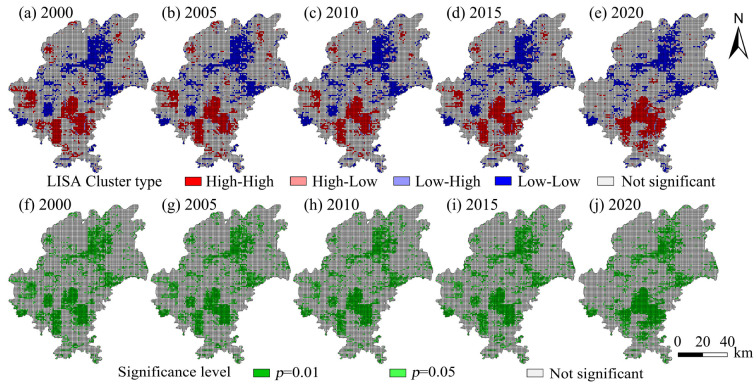
LISA clustering analysis and significance level of human activity intensity and habitat quality changes in Guiyang from 2000 to 2020.

**Figure 7 ijerph-19-14294-f007:**
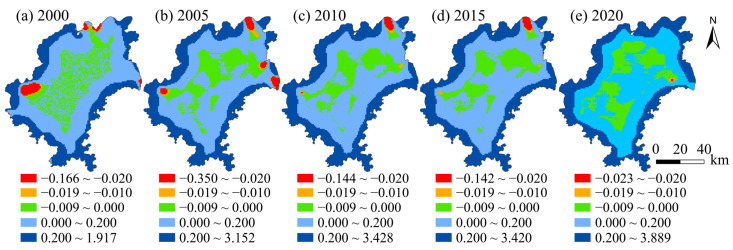
Spatial distribution map of regression coefficients of human footprint index in Guiyang from 2000 to 2020.

**Table 1 ijerph-19-14294-t001:** Maximum influenced distance and weight of threat factors.

Threat Factor	Maximum Distance (km)	Weight	Decay Type
Cultivated land	4	0.6	Linear
Construction land	8	0.4	Exponential
Unutilized land	6	0.5	Linear

**Table 2 ijerph-19-14294-t002:** Habitat suitability of land-use types and relative sensitivity to various threat factors.

Land-Use Type	Habitat Suitability		Threat Factor	
		Cultivated Land	Construction Land	Unutilized Land
Cultivated land	0.3	0	0.5	0.3
Forestland	1.0	0.8	1	0.4
Grassland	0.9	0.8	0.6	0.3
Water body	0.7	0.6	0.7	0.3
Construction land	0.0	0.6	0.9	0.3
Unutilized land	0.5	0	0	0

**Table 3 ijerph-19-14294-t003:** Human activity intensity coefficient of different land-use types.

Parameter	Grassland	Forestland	Cultivated Land	Unutilized Land	Reservoir and Pond	Construction Land
Lohani	0.09	0.12	0.61	0.05	0.33	0.96
Leopold	0.08	0.14	0.59	0.07	0.29	0.94
Delphi	0.09	0.13	0.64	0.08	0.35	0.96
Average value	0.09	0.14	0.61	0.07	0.32	0.95

**Table 4 ijerph-19-14294-t004:** Area and change proportion of habitat quality grade in Guiyang from 2000 to 2020.

Habitat Quality Grade	2000 Year		2005 Year		2010 Year		2015 Year		2020 Year	
	Area/km^2^	Proportion%	Area/km^2^	Proportion%	Area/km^2^	Proportion%	Area/km^2^	Proportion%	Area/km^2^	Proportion%
Low	190.90	2.37	225.86	2.81	251.68	3.13	290.95	3.66	559.26	6.95
Relatively low	2565.49	31.90	2739.62	34.06	2578.42	32.06	2529.74	31.81	2487.48	30.93
Moderate	831.04	10.33	929.39	11.56	813.57	10.12	816.74	10.27	831.13	10.33
Relatively high	1406.02	17.48	1396.46	17.36	1368.57	17.02	1371.45	17.25	1456.94	18.11
High	3049.34	37.91	2751.47	34.21	3030.56	37.68	2943.22	37.01	2707.99	33.67

## Data Availability

Land-use change data were downloaded from the Resources and Environment Data Sharing Center of Chinese Academy of Sciences (http://www.resdc.cn), (accessed on 1 January 2020). economic development data from the statistical yearbook published by the People’s Government of Guiyang(http://www.guiyang.gov.cn), (accessed on 1 January 2020). The base map derived from the geographic information public service platform (https://guizhou.tianditu.gov.cn), (accessed on 1 January 2020).

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
