# Peer review of "Correlation between Spatio-Temporal Evolution of Habitat Quality and Human Activity Intensity in Typical Mountain Cities: A Case Study of Guiyang City, China"

_ijerph, 2022, doi:10.3390/ijerph192114294_

Round 1

Reviewer 1 Report

On the basis of land use and related data from 2000 to 2020, the author made a scientific and rigorous analysis of human activity intensity and habitat quality in Guiyang by using InVEST, binary autocorrelation analysis and geographically weighted regression model, focusing on the impact of human activity intensity on human settlement quality and its correlation. It provided some reference for the study of ecological environment and sustainability of mountain city. After reviewing the whole manuscript, I think it has high academic logic, complete and substantial content and beautiful pictures. However, there are still some areas to be improved in the content details and discussion. The specific problems are as follows. 

1. In this paper, the author had written "Based on the land use data of Guiyang in 2000, 2010 and 2020, ..." (Line 11) Why did the last paragraph of the introduction became "Based on the land use change data of 2000, 2005, 2010, 2015 and 2020, ..." (line 93)? From the perspective of the whole paper, the basic data of the authors should be the land use data of 2000, 2005, 2010, 2015 and 2020. In addition, before the introduction, the author wrote "InVEST model", and after the introduction, the author wrote "INVEST model". Please be sure to keep the contents of the manuscript and pictures consistent. 

2. In 2.1 Overview of the study area, I suggest that the author convert industrial value, per capita GDP and financial revenue into an international currency unit, such as USD 9999 billion, so as to facilitate the reading of international readers. On line 118, please indicate "The total land area of the city is 8,043km²..." What year is it? At the end of the paragraph, the order of land area introduction is rearranged to be consistent with that in Table 2. 

3. In Figure 1, "Elevation" lacked the necessary units of measurement. 

4. In Table 4, the author is requested to ensure the correct expression of the measurement unit, whether it is "km2" or "km2".

5. In 4.2, author is requested to move the description (HAI) of the “human activity intensity index” to the place where it first appears in the manuscript. And be careful not to repeat the instructions. 

6. In Figure 7, the legend of 2020 from 0.001 to 0.200 does not correspond to the expression in the figure. 

7. 5. Discussion part is well written, but mostly focuses on the practical significance of the author's research results for Guiyang. As an academic paper, 5. Discussion lacks the description of theoretical significance. I suggest that the author add theoretical innovations to the research results in the discussion and compare them with other related studies. What is the difference between the author's research and other related studies? What improvements have been made?

Author Response

Comments and Suggestions for Authors 1

On the basis of land use and related data from 2000 to 2020, the author made a scientific and rigorous analysis of human activity intensity and habitat quality in Guiyang by using InVEST, binary autocorrelation analysis and geographically weighted regression model, focusing on the impact of human activity intensity on human settlement quality and its correlation. It provided some reference for the study of ecological environment and sustainability of mountain city. After reviewing the whole manuscript, I think it has high academic logic, complete and substantial content and beautiful pictures. However, there are still some areas to be improved in the content details and discussion. The specific problems are as follows.

Response to Reviewer 1:

Thank you very much for your comments and advices on our manuscript. Your comments are very helpful for us to improve our paper. According to your comments, the paper has been revised carefully. The following contents are the detailed response to your every comment and the explanations of the questions you have mentioned. Meanwhile, all the questions you mentioned have been revised at the according content in the resubmitted paper.

  1. In this paper, the author had written "Based on the land use data of Guiyang in 2000, 2010 and 2020, ..." (Line 11) Why did the last paragraph of the introduction became "Based on the land use change data of 2000, 2005, 2010, 2015 and 2020, ..." (line 93)? From the perspective of the whole paper, the basic data of the authors should be the land use data of 2000, 2005, 2010, 2015 and 2020. In addition, before the introduction, the author wrote "InVEST model", and after the introduction, the author wrote "INVEST model". Please be sure to keep the contents of the manuscript and pictures consistent.

Response to Reviewer(s)' Comments:

Thank you very much for your comments and advice on our manuscript. We have now carefully checked the usage of proper nouns and detailed expression of the manuscript according to your suggestion.

  1. In 2.1 Overview of the study area, I suggest that the author convert industrial value, per capita GDP and financial revenue into an international currency unit, such as USD 9999 billion, so as to facilitate the reading of international readers. On line 118, please indicate "The total land area of the city is 8,043km²..." What year is it? At the end of the paragraph, the order of land area introduction is rearranged to be consistent with that in Table 2.

Response to Reviewer(s)' Comments:

Thank you. According to your comments, the questions you mentioned have been revised at the according content in the resubmitted paper.

  1. In Figure 1, "Elevation" lacked the necessary units of measurement.

Response to Reviewer(s)' Comments:

Thank you. According to your comments, We have added the Elevation measuring units m.

  1. In Table 4, the author is requested to ensure the correct expression of the measurement unit, whether it is "km2" or "km2".

Response to Reviewer(s)' Comments:

Thank you. We have been revised at the according content in the resubmitted paper.

  1. In 4.2, author is requested to move the description (HAI) of the “human activity intensity index” to the place where it first appears in the manuscript. And be careful not to repeat the instructions.

Response to Reviewer(s)' Comments:

Thank you for the suggestion. We have moved the description (HAI) of the “human activity intensity index” to the place where it first appears in the manuscript.

  1. In Figure 7, the legend of 2020 from 0.001 to 0.200 does not correspond to the expression in the figure.

Response to Reviewer(s)' Comments:

Thank you. We have been revised at the according content in the resubmitted paper.

  1. 5. Discussion part is well written, but mostly focuses on the practical significance of the author's research results for Guiyang. As an academic paper, 5. Discussion lacks the description of theoretical significance. I suggest that the author add theoretical innovations to the research results in the discussion and compare them with other related studies. What is the difference between the author's research and other related studies? What improvements have been made?

Response to Reviewer(s)' Comments:

Thank you. We have been revised at the discussion in the resubmitted paper.

Previous studies on the quality of human settlements in Guiyang are mainly based on the evaluation of a single habitat quality unit in the city, and lack of comprehensive consideration of the spatio-temporal coupling relationship between habitat quality changes and human activity intensity. This paper innovatively proposes a comprehensive regional factor index for evaluating urban habitat quality assessment and land use. This index comprehensively considers the correlation between habitat quality change and human activity intensity, and reveals the spatio-temporal characteristics of land use, human activity intensity and urban spatial pattern evolution.

Sincerely yours,

Yongfei LUAN and Guohe HUANG, et al.

2022-10-25.

Reviewer 2 Report

The paper entitled Correlation between spatio-temporal evolution of habitat quality and human activity intensity in typical mountain cities: A case study of Guiyang City, China is analysed.

Overall, the article has a good scientific coherence. The account is clear and orderly. In relation to this, it is suggested to go through the whole paper and gather in the methods section the information related to the methods. It is a well thought out section but there are references to methods in the results section that should be specified in the methods section.

Questions and suggestions to be taken into account:

Introduction: it is suggested to contextualise what is meant by a mountainous city and whether there is any classification in China to determine when a city falls into this category.

Data Material Sources and Research Methods: this section is described thoroughly and in great detail. Overall, it is well thought out and sufficiently clear.

Line 108-125: Review the units described in the paragraph and their punctuation. It is suggested to include in the description of the area the maximum and minimum altitude of the province.

Line 179: Table 1 and 2: suggest capitalising the first letter of "unutilized land".

Results:

Line 240-242: it is suggested to use the same name for the intervals as in table 4.

Line 245: it is suggested to increase the quality of figure 3, especially sub-figure f.

Line 270-295: it would be of great interest to clarify and increase the knowledge of the research to explain the possible causes of the fluctuations experienced by the HAI index in some sectors of the studied area. Some of these causes are described but not in all cases. It may be of interest to include a table that brings together and classifies each sector with the process and the cause.

Line 299-303. It is suggested to move the information on the methods used in this section to the methods section and to explain in detail their relevance and development.

Section: 4.3. Bivariate spatial correlation of human activity intensity. In this section there are also allusions to methodological issues that have not been pointed out in the corresponding section.

Line 335: it is suggested to include in the figure the year to which each of the graphs belongs.

Discussion

The discussion of the results is generally correct. The discourse and discussion could be broadened to include topics already raised in the introduction, such as: conservation of areas with high biodiversity versus urbanisation processes, planning and land use policies, etc. This would enrich the final part of the paper, which is perhaps not very closed. In addition, it is suggested that this section should include results and quotes from other studies that can compare the processes and quality of life indices obtained in areas with the same geographical and socio-economic characteristics: large cities in mountain areas. This issue is not discussed very much.

Conclusions

Some questions remain unaddressed, such as the causes of changes in land use, most probably induced and poorly controlled by spatial planning policies, and also the intense territorial polarisation which occurs and increases throughout the period studied. 

Author Response

Comments and Suggestions for Authors 2

The paper entitled Correlation between spatio-temporal evolution of habitat quality and human activity intensity in typical mountain cities: A case study of Guiyang City, China is analysed.

Overall, the article has a good scientific coherence. The account is clear and orderly. In relation to this, it is suggested to go through the whole paper and gather in the methods section the information related to the methods. It is a well thought out section but there are references to methods in the results section that should be specified in the methods section.

Response to Reviewer 2:

Thank you very much for your comments and advices on our manuscript. Your comments are very helpful for us to improve our paper. According to your comments, the paper has been revised carefully. The following contents are the detailed response to your every comment and the explanations of the questions you have mentioned. Meanwhile, all the questions you mentioned have been revised at the according content in the resubmitted paper.

Questions and suggestions to be taken into account:

Introduction: it is suggested to contextualise what is meant by a mountainous city and whether there is any classification in China to determine when a city falls into this category.

Response to Reviewer(s)' Comments:

Thank you. Mountainous city is a generalized concept. mountainous, including geographically divided mountains, hills and rugged plateaus, account for about 69% of the country's land area. mountainous city refers to cities mainly distributed in the above mountainous regions, forming urban forms and habitats that are completely different from plain areas. At present, there is no clear definition of mountain city. However, since 2013, Guizhou Province has initially formed a new concept of urban development with Guizhou characteristics, and set out on a road of urban complex development with Guizhou mountainous city characteristics.

Data Material Sources and Research Methods: this section is described thoroughly and in great detail. Overall, it is well thought out and sufficiently clear.

Line 108-125: Review the units described in the paragraph and their punctuation. It is suggested to include in the description of the area the maximum and minimum altitude of the province.

Response to Reviewer(s)' Comments:

The highest elevation of the province is 2885m and the lowest is 152m.

Line 179: Table 1 and 2: suggest capitalising the first letter of "unutilized land".

Response to Reviewer(s)' Comments:

Thank you. We have been revised at the according content in the resubmitted paper.

Results:

Line 240-242: it is suggested to use the same name for the intervals as in table 4.

Response to Reviewer(s)' Comments:

Thank you. We have been revised at the according content in the resubmitted paper.

Line 245: it is suggested to increase the quality of figure 3, especially sub-figure f.

Response to Reviewer(s)' Comments:

Thank you. We have been revised at the according content in the resubmitted paper.

Line 270-295: it would be of great interest to clarify and increase the knowledge of the research to explain the possible causes of the fluctuations experienced by the HAI index in some sectors of the studied area. Some of these causes are described but not in all cases. It may be of interest to include a table that brings together and classifies each sector with the process and the cause.

Response to Reviewer(s)' Comments:

We quite agree with your opinion, but at present, we are limited by certain factors in obtaining data, so we cannot collect specific data comprehensively. However, it is a factor that should be taken into consideration in our future research, and we will try to improve it in the future research.

Line 299-303. It is suggested to move the information on the methods used in this section to the methods section and to explain in detail their relevance and development.

Response to Reviewer(s)' Comments:

Thank you. We have been revised at the according content in the resubmitted paper.

Section: 4.3. Bivariate spatial correlation of human activity intensity. In this section there are also allusions to methodological issues that have not been pointed out in the corresponding section.

Line 335: it is suggested to include in the figure the year to which each of the graphs belongs.

Response to Reviewer(s)' Comments:

Thank you. We have been revised at the according content in the resubmitted paper.

Discussion

The discussion of the results is generally correct. The discourse and discussion could be broadened to include topics already raised in the introduction, such as: conservation of areas with high biodiversity versus urbanisation processes, planning and land use policies, etc. This would enrich the final part of the paper, which is perhaps not very closed. In addition, it is suggested that this section should include results and quotes from other studies that can compare the processes and quality of life indices obtained in areas with the same geographical and socio-economic characteristics: large cities in mountain areas. This issue is not discussed very much.

Response to Reviewer(s)' Comments:

Thank you. We have been revised at the according content in the resubmitted paper.

Conclusions

Some questions remain unaddressed, such as the causes of changes in land use, most probably induced and poorly controlled by spatial planning policies, and also the intense territorial polarisation which occurs and increases throughout the period studied. 

Response to Reviewer(s)' Comments:

Thank you. Due to the urban development in recent years, urban land sprawl and construction land demand increase, which is affected by the blindness of land use and the inadequate implementation of land space planning and management.

We highly appreciate the reviewers and editor’s kind efforts, comments and suggestions helped to improve our research work as well as our manuscript.

Thank you very much again.

Sincerely yours,

Yongfei LUAN and Guohe HUANG, et al.

2022-10-25.